# Using Eye Tracking to Explore the Impacts of Geography Courses on Map-based Spatial Ability

**Weihua Dong [1], Qi Ying [1,*], Yang Yang [1], Siliang Tang [1,*], Zhicheng Zhan [1], Bing Liu [1] and Liqiu Meng [2]**

[1] Beijing Key Laboratory of Environmental Remote Sensing and Digital Cities, Research Centre of Geospatial Cognition and Visual Analytics and Faculty of Geographical Science, Beijing Normal University, Beijing 100875, China; dongweihua@bnu.edu.cn (W.D.); yangyang1229@mai.bnu.edu.cn (Y.Y.); zhanzhicheng@mail.bnu.edu.cn (Z.Z.); liubing_geo@mail.bnu.edu.cn (B.L.)

[2] Technical University of Munich, 80333 Munich, Germany; liqiu.meng@tum.de

* Correspondence: yingqi@mail.bnu.edu.cn (Q.Y.); jean_tsl@foxmail.com (S.T.)

**Abstract:** As part of geography education, geography courses play an important role in the development of spatial ability. However, how geography courses affect map-based spatial ability has not been well documented. In this study, we use an eye-tracking method to explore the impact of geography courses on map-based spatial ability. We recruited 55 undergraduates from Beijing Normal University (BNU) to attend the map-based spatial ability test before and after six-month geography courses arranged by the Faculty of Geographical Science, BNU. The results show that the participants' map-based spatial ability significantly improved after taking the geography courses; specifically, accuracy increased by 22.3% and response time decreased by 14.7% after training. We analysed two types of eye-movement behaviour; in terms of processing measures, the fixation duration of the topographic map decreased by 18.4% and the fixation distribution was more concentrated after training, and in terms of matching measures, participants have more switch times per second for both photographed scenes and topographic maps. Switch times between options decreased by 48.2%, which is a notable decrease. These empirical results are helpful for the design of geography courses that improve map-based spatial ability.

**Keywords:** eye tracking; spatial ability; geography education; geography courses

## 1. Introduction

Spatial ability is a fundamental ability used in daily life. Equipped with spatial ability, people could be quickly aware of an unfamiliar environment, acquire key environment information effectively and more easily put forward sustainable and realistic solutions to some spatial-based social issues and environmental problems. The definition of spatial ability has been continually argued over the past several decades [1], but it is recognized to be related to skills that involve the retrieval, retention, and transformation of visual information in spatial contexts [2]. Moreover, it is generally accepted that spatial ability is composed of several distinct but interrelated factors. Spatial orientation ability and spatial visualization ability are two primary abilities that are widely included in the definition of spatial ability. Spatial orientation skill refers to the ability to imagine oneself or a configuration from different perspectives [3], while spatial visualization ability is the ability to mentally restructure or manipulate the components of visual stimuli and involves recognizing, retaining, and recalling configurations when the figure or parts are moved [4]. Some map-based tasks, such as wayfinding and self-orientation, require people to finish a task by connecting a map with the surrounding environment. In the last few decades, there has been a surge of interest in the study of map-based spatial ability. Map-based spatial

ability includes not only basic map reading skills but also critical map literacy. The latter involves possessing the knowledge and skills that enable analysis and evaluation of a map [5].

Previous studies have investigated individual differences in spatial ability. Some natural human attributes, such as gender and age, have been proven to affect spatial ability. Among studies focusing on gender differences in spatial ability, Linn et al. (1985), Voyer et al. (1995), Weckbacher and Okamoto (2014) provided evidence that men outperform women in spatial rotation tasks [6–8]. However, Stumpf and Eliot's study (1995) revealed a female advantage on tasks involving visual memory [9]. For the influence of age on spatial ability, Harris et al. and Nardini et al. confirmed that spatial skills were improving throughout childhood [10,11]. Over an adult's lifespan, Borella et al. proved that the effect of age on spatial ability is considerable [12]. Social factors, according to some research, also play an important role in individual differences in terms of spatial ability. Many studies indicated that educational background affects spatial ability; some studies on college students support this indication. According to Yoon and Mann, undergraduate students who major in science, technology, engineering and mathematics (STEM) outperformed those of non-STEM majors in terms of spatial ability when other conditions were equivalent [13]. Titus et al. found that the spatial ability of engineering majors that rely heavily on spatial ability skills improved more than that of other majors over the course of undergraduate studies [14]. According to the aforementioned studies, educational background affects spatial ability mainly due to access to different knowledge, strategies and skills.

Geography education is a proven way to develop map-based spatial ability. Dong et al. compared geographers and non-geographers according to topographic map-reading test performance and found that geographers have an advantage in the effectiveness and efficiency of reading topographic maps to solve geographical problems [15]. For the influence of geography courses on spatial ability, Titus and Horsman evaluated students' spatial skills twice—before and after a geology course—and the results suggested that spatial abilities can improve through the practice provided in a geology course [16]. It is argued that students benefit from geography education because it sharpens their cognitive mapping and spatial thinking skills [17]. Geography education also helps students accumulate expertise, which is important in map-based tasks. Ooms, et al. found that expert map users are significantly more efficient in terms of visual search performance when shown basic screen maps [18] and topographic screen maps [19]. However, Burian et al. found that even expert users can be misled by the incorrect use of map symbols and thus reduce their accuracy [20].

Geography education improves spatial ability through teaching time, materials and methods. Teaching duration and teaching materials are factors that influence the level of spatial ability. The level of spatial ability is positively correlated with the duration of geography education and the level of the learning materials. Towle et al. used the rotation tests used to measure the spatial ability of students—and compared the spatial ability scores of the upper and lower grades. The results indicated that students of higher grades scored higher [21]. In the assessment of students' spatial abilities by Titus and Horsman (2009), students in higher-level courses had better-developed visualization skills than those in introductory courses [16]. In addition, spatial ability improvement was affected by the geography teaching method. The geographic information system (GIS) is regarded as a useful tool for improving students' spatial ability [22]. Murray proved that providing GIS in university teaching is beneficial for constructive learning in geography courses [23]. According to Cheung et al., geography courses using GIS and satellite remote sensing promote the spatial ability of students and enhance the understanding of spatial patterns and relationships [24]. Some geography courses have incorporated VR (virtual reality) or AR (augmented reality) [25,26]. Students of different spatial abilities will benefit differently from interactive 3D animations and simulations [27,28]. Therefore, we know that the quality and method of geography education can have a profound impact on the improvement of students' spatial ability. However, more research is necessary to address why these improvements in spatial ability occur.

For individuals, advances in spatial abilities are significant. Spatial ability is closely related to spatial intelligence, which is essential when people engage in artistic, scientific, mathematical or even literary activities. The importance of spatial ability in educational pursuits and the world

of work has been proved, with particular attention devoted to STEM domains [29]. A number of studies underscore the importance of spatial ability for accomplishments in STEM disciplines [29–31]. Therefore, for students majoring in geography, advances in spatial ability help them to improve the competency in interdisciplinary work. With the enhancement of map-based spatial ability, students' critical map literacy improves, which is closely related to their critical thinking. Both the critical thinking and the competence in interdisciplinary work are parts of key competencies in sustainability [32]. In Germany, developing these key competencies has been discussed as part of the central educational objective of ESD (Education for Sustainable Development) [33].

Accurately measuring spatial ability is key to detecting changes in spatial ability. Previous studies developed many approaches to assess spatial ability, such as questionnaires, written tests and online tests. There are some classic tests, including the Vandenberg Mental Rotations Test [34], the Santa Barbara Sense-of-Direction Scale [35] and four spatial tests from the Kit of Factor-Referenced Cognitive Tests: The card rotations test, the cube comparisons test, the paper folding test, and the surface development test [36,37]. Most of these tests are used to evaluate the spatial ability of subjects via test score or response time, and they pay more attention to the test results than the exploration of the thinking process.

The emergence and popularization of eye-tracking technology have brought about new possibilities for the quantitative study of visual information processing. There are two basic components of eye movement: Fixation (the position on a screen where the eye pauses for a certain period) and saccade (the rapid movement between fixations) [38]. Many studies on spatial ability and spatial thinking employed eye tracking. By combining saccade and fixation durations, Chen et al. examined the effectiveness of the spatial ability of high school students [39]. Based on visual and statistical analyses of eye-movement recordings, Ooms et al. explored the difference in the map memorization abilities of geographers and non-geographers [40]. Dong et al. used gaze series parameters (fixation count, fixation frequency) to analyse the differences between geographers and non-geographers in terms of spatial ability regarding topographic maps [15]. Thus, eye tracking is an effective way to study the differences in spatial ability before and after geography courses.

In this study, we used an eye-tracking method to evaluate the map-based spatial ability of undergraduate students in geographic specialties. By comparing the students' performances before and after 6-month geography courses, we determined how these courses affected map-based spatial ability. We evaluated the test performances of participants in terms of accuracy and task completion time. Additionally, we analysed cognitive and information search processes through eye-movement data. This study provides a quantitative assessment and data comparison of map-based spatial ability and evaluated the influence of geography courses on the map-based spatial ability of participants. Therefore, this study provides necessary evidence collection toward the data-driven exploration of spatial ability.

## 2. Methods

### 2.1. Participants

Fifty-five undergraduate students (30 females; 19 ± 2 years old) majoring in geography from Beijing Normal University (BNU) participated in this experiment. The experiment was reviewed and approved by the local institutional review board (IRB). All participants provided their written informed consent. The participant sample rates (calculated by dividing the eye-tracking samples that were correctly identified by the number of attempts) of the two tests were above 70%.

Before the first test, participants had not taken any geography courses. For the second test, the participants had completed geography courses that were approximately six months in length. The geography courses included Cartography, Physical Geography and Practice of Physical Geography. The geography courses included two parts: Classroom learning and field training. Cartography and Physical Geography, which were held twice per week (three hours per week), are classroom learning

courses that are taken in parallel Practice of Physical Geography is a field-training course. Classroom learning includes map projections, coordinate systems and topographic maps, which help improve the spatial orientation ability for surveying and basic mapping knowledge. Field training included a five-day comprehensive exercise and four one-day short-distance field training exercises. Through field training, the students were instructed on the basic methods of field observation, recording, measurement, and location and orientation determination in a field environment.

*2.2. Apparatus*

The experiment used a Tobii T120 eye tracker with a 22-inch monitor and a sampling rate of 60 Hz. The instrument's recording accuracy was $0.5°$, and the error was less than or equal to $0.1°$. The spatial resolution was $0.2°$, and the head movement error was within $0.2°$. The topographic map was displayed on a screen with a resolution of $1280 \times 1024$ pixels. In addition, the laboratory had good lighting and no disturbances.

*2.3. Materials*

We developed an interface for a map-based spatial cognition test (Figure 1) using C#. The upper-left section of the interface shows a photograph generated by Google Earth. Under the image, we set four buttons for participants to use to select answers, confirm answers, switch to the next task and submit answers. The right side of the interface shows a topographic map of Lushan Mountain that was produced using a 30-meter-resolution digital elevation model (DEM) from Geospatial Data Cloud (http://www.gscloud.cn/) in ArcMap. On the topographic map, some candidate positions or directions were given in the form of red points or arrows. The participant's task was to complete the positioning and orientation tasks according to the photographs and red cues using the topographic map. Participants were able to complete the task by clicking the four buttons provided on the interface. Participants could click the 'Candidate' button to select the optional point and direction. After selecting an answer, clicking the 'Confirm' button would confirm the answer and clicking the 'Next task' button switched to the next task. Participants had to click the 'Submit' button when all tasks were finished.

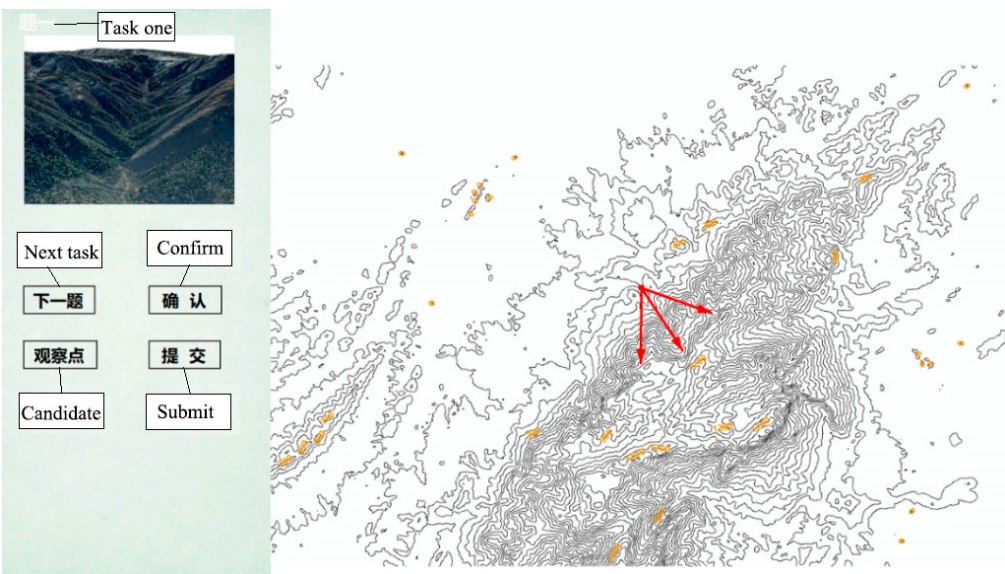

**Figure 1.** Task interface of the experiment (task one).

*2.4. Procedures*

For tasks one through seven, the topographic map of Lushan Mountain was displayed on the right side of the screen. There were discrete points and several directions for each point on the topographic map. On the left side of the interface, a corresponding photograph was presented. For each task,

the participants had to match the location from the photograph to the corresponding location on the topographic map. For tasks eight through ten, only discrete candidate points were displayed on the right side of the screen. Participants had to select one discrete point on the topographic map and then draw directions based on the selected point.

In addition, to compare the changes in the map-based spatial ability of the participants before and after the geography courses, we tested the same group of participants before and after training. They were not told the right answers for each task after the first test, and the order of ten tasks was rearranged in the second test. A single test included two parts. First, we introduced the test to the participants to ensure they were familiar with the interface and understood the task. Then, the test officially began. Ten tasks appeared in sequence on the interface. There were 6 types of tasks. (Table 1)

**Table 1.** Task list.

| Task | Description | Interface |
|---|---|---|
| Task 1 | Three alternative directions from the same site |  |
| Task 2<br>Task 3 | Two alternative directions from different sites |  |
| Task 4<br>Task 5 | Three alternative sites with two alternative directions from each site |  |
| Task 6<br>Task 7 | Three alternative sites with three alternative directions for each site |  |
| Task 8<br>Task 9 | Given one site, the participants marked their own direction |  |
| Task 10 | Given two alternative points, the participants marked their own direction |  |

## 2.5. Analysis

To compare the map-based spatial abilities of the participants before and after the geography courses, we used the general performance to evaluate efficiency and accuracy. In addition, we used eye-tracking data to analyse the cognitive process of the participants. We conducted a series of qualitative and quantitative analyses of the data obtained. As shown in Table 2, the eye-tracking data included processing measures, search measures and other metrics.

**Table 2.** Metrics for quantitative analysis.

| Types | Metrics (Unit) | Interpretation |
|---|---|---|
| General performance | Response time (seconds) | Time needed to complete the task |
| | Total score | Number of correct judgements (with regard to position and direction) over ten tasks. |
| Processing measures | Total fixation count (count) | Number of fixations within an AOI[1] |
| | Total fixation duration (seconds) | All durations of fixation on one AOI |
| | Average fixation duration (seconds) | Average of all durations of fixation on one AOI |
| Matching measures | Switch times (count) | Number of switches between the photograph and the topographic map or between different options within one task |
| | Switch times per second (count) | Number of switches per second between the photograph and the topographic map or between different options within one task |

Note: AOI is used as an abbreviation for "area(s) of interest".

The qualitative analysis includes the use of gaze parameters (i.e., fixation and saccade) to draw fixation HexBin plots (Figure S1 to Figure S10) for the participants, through which we could analyse the cognitive process of each participant's readings.

The general performance included response time and total score. Response time is the time taken to complete all tasks. For each task, if the answer was correct, then the participants received 1 point; if the wrong answer (i.e., the wrong point and/or the wrong direction) was given, then the participants received no points. Total score is a simple sum of the points granted for each task; there were ten tasks in total. For the last three tasks, directions within $10°$ on both sides of the correct direction were considered valid answers.

The processing measures consisted of total fixation count, total fixation duration, and average fixation duration, which reflected the ability of the participants to process information. Total fixation count refers to the fixation count in the area of interest (AOI). A higher fixation count mean more attention was given to that area. A longer total fixation duration means that the participants were more concerned with the AOI, which corresponded to more difficult-to-process information and may be due to the participants' increased interest in the stimulating materials [41]. Studies show that fixation duration can reflect the degree of cognitive difficulty in a region [42].

The number of switch times was equal to the number of times the participant's fixation shifted between the photographed scene and the topographic map, which reflected the participant's spatial memory and information processing efficiency. Similarly, within a task, fixation shifts between different options were also considered. On this basis, the number of fixation switches (between the topographic map, the photographed scene, and between different options of the same task) was calculated per unit of time. This indicator reflected the efficiency of the participants' gaze shifts across the different areas of the map.

By considering that the essential information is distributed throughout the photographed scene and topographic map, we choose to statistically analyse the photographed scene and topographic map as the AOI.

## 3. Results

We wanted to explore whether there was a significant difference in the participants' performance before and after the test. In this case, the paired sample t-test was considered. However, three hypotheses need to be considered to perform paired sample t-test:

**H1.** *The dependent variable is a continuous variable.*

**H2.** *There are no significant outliers in the difference of dependent variables between two correlated (paired) groups.*

**H3.** *The difference of dependent variables between two correlated (paired) groups approximately obeys a normal distribution.*

If the above hypotheses cannot be completely satisfied, the Wilcoxon symbolic rank test was adopted.

## 3.1. General Performance

Figure 2 shows that the total score of the participants after taking geography courses (mean and standard deviation of M = 5.55 and SD = 1.53, respectively) were higher than those before training (M = 4.31, SD = 1.50). The results of the paired t-test showed that there was a significant difference in the accuracy of the results before and after training (95% CI: 0.70 to 1.77, P = 0.000, <0.001) (Figure 2a). After training, the RT (M = 539.90 s, SD = 218.54) was higher than that before training (M = 460.44 s, SD = 198.82). The paired t-test results showed that there was a notable decrease after training (P = 0.025, <0.05) (Figure 2b). These results showed that, after training, the participants could solve problems more quickly and accurately.

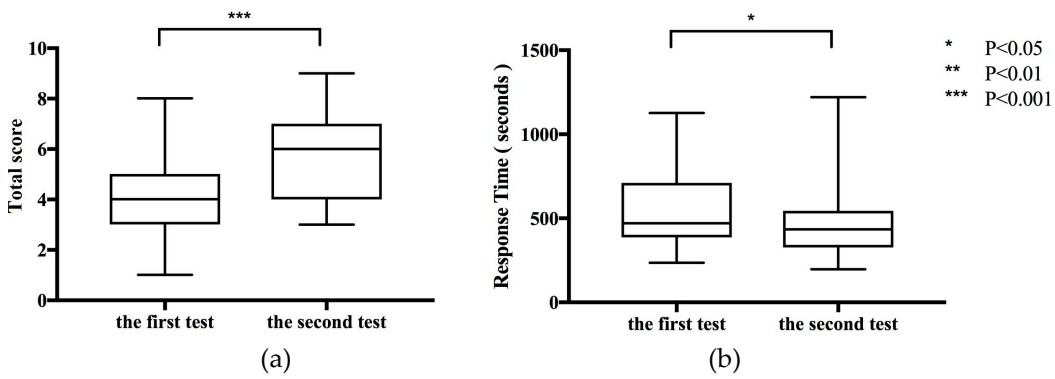

**Figure 2.** Statistics for the general performances before and after training and the significance test results. (**a**) Total score ($-1.24 \pm 1.72$, t = $-4.659$, p = 0.000); (**b**) Response time ($75.46 \pm 209.96$, t = 2.329, p = 0.025).

## 3.2. Processing Measure

Figure 3a,b show that, in the first test, participants had a higher total fixation count for both the photographed scene (M = 305.40, SD = 121.2) and the topographic map (M = 757.96, SD = 331.24) than in the second test (M = 269.10, SD = 92.26 and M = 656.78, SD = 243.31 for the photographed scene and the topographic map, respectively). However, the results of paired sample t-test showed that there was no significant difference in the mean of the above fixation counts before and after training.

In terms of total fixation duration (Figure 3c,d), participants had a higher total fixation duration on the photographed scenes in the first test (M = 70.41, SD = 27.84) than in the second test (M = 61.24, SD = 22.53). Moreover, the total fixation duration on the topographic maps in the second test (M = 181.06, SD = 64.49) was lower than that in the first test (M = 221.90, SD = 119.29). Since the difference of fixation duration in the topographic map was not normally distributed, that is, it did not meet hypothesis 3, the Wilcoxon symbolic rank test was carried out (confidence level: 95%). And the result indicated that the difference in median between the two groups of data was statistically significant.

However, as shown in Figure 3e,f, the average fixation duration for the photographed scenes in the second test was slightly higher than that for the first test, whereas the average fixation duration for

the topographic maps in the first test was higher than that for the second test; however, both showed no significant difference.

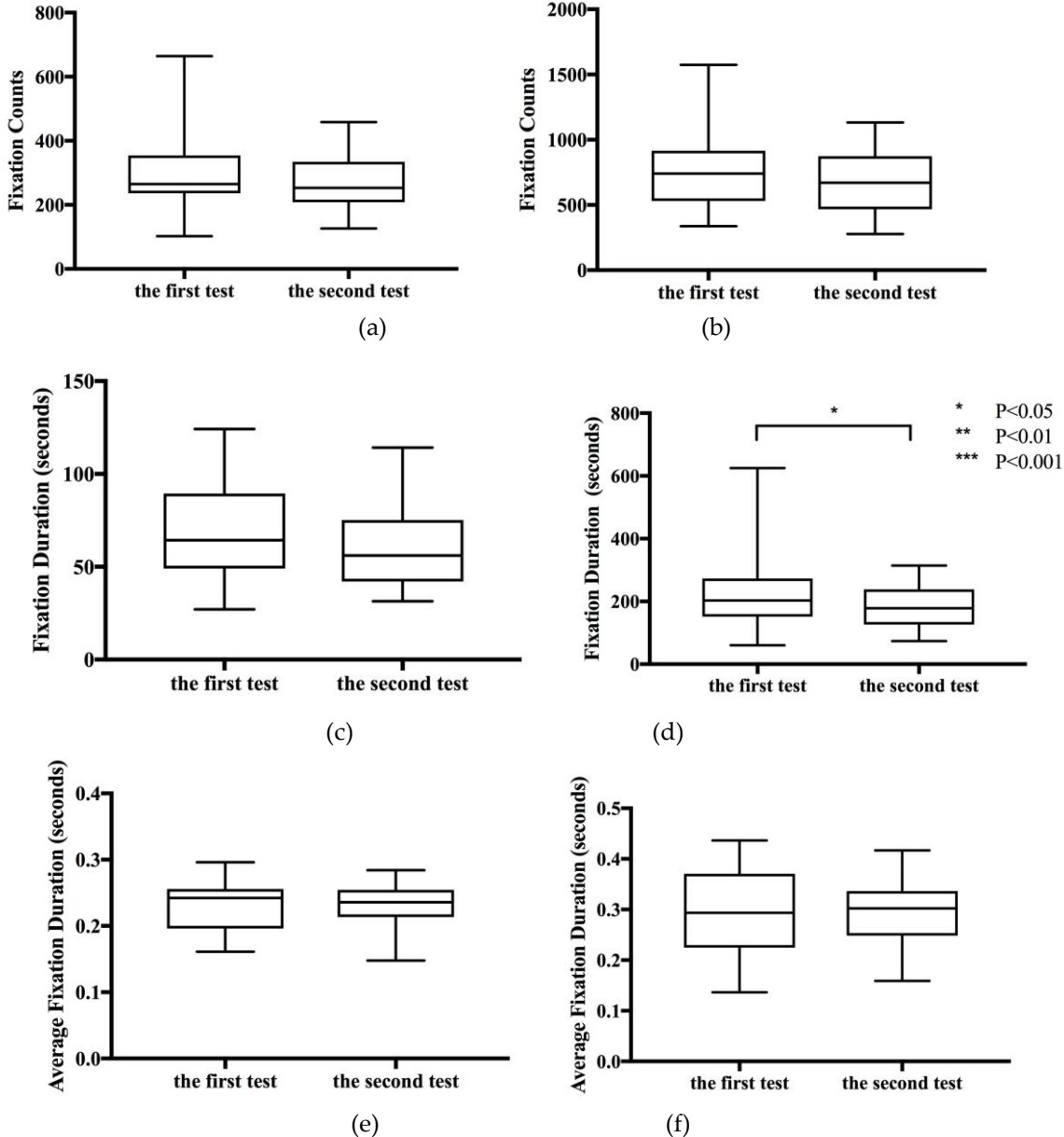

**Figure 3.** Statistical comparison of total fixation count, total fixation duration, and average fixation duration before and after training. (**a**) total fixation count of the photographed scene (36.32 ± 118.22, t = 1.536, p = 0.138); (**b**) total fixation count of the topographic map (101.20 ± 296.19, t = 1.714, p = 0.099); (**c**) total fixation duration of the photographed scene (9.17 ± 28.52, t = 1.608, p = 0.121); (**d**) total fixation duration of the topographic map (Nonparametric test result: z = −2.085, p = 0.037); (**e**) average fixation duration of the photographed scene (0.00 ± 0.04, t = 0.412, p = 0.684); (**f**) average fixation duration of the topographic map (0.01 ± 0.07, t = 0.601, p = 0.553).

*3.3. Matching Measures*

3.3.1. Number of Switches between the Photographed Scene and the Topographic Map

Based on the coordinates and sequence information of the fixation point, we calculated the switch times between different areas on the answer interface. The results of switch times for the photographed

scene and the topographic map are shown in Figure 4. The results of the paired t-test (confidence level: 95%) indicated that there was no notable difference in the number of switch times between these two regions. However, number of switches per unit time increased significantly after training.

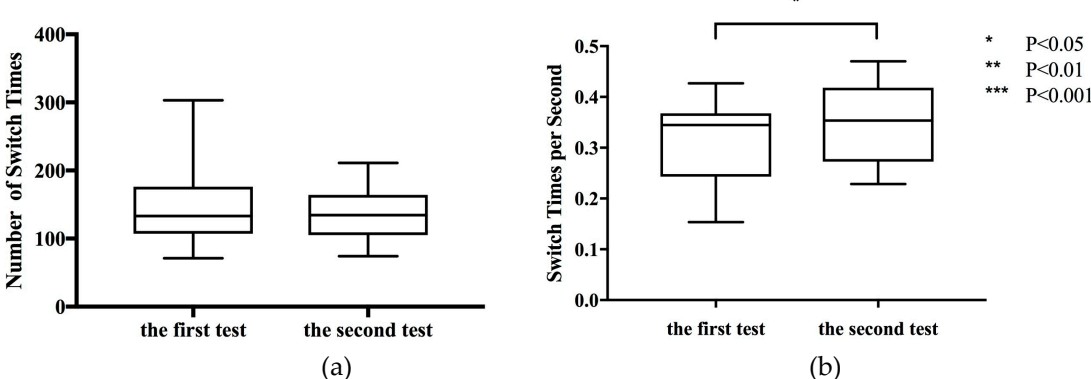

**Figure 4.** Statistics of switch times and switch times per second for the photographed scene and the topographic map before and after training and the results of the significance test. (**a**) switch times (11.58 ± 60.48, t = 0.938, p = 0.358); (**b**) switch times per second (-0.03 ± 0.07, t = -2.250, p = 0.358).

### 3.3.2. Switch Times between Different Options in One Task

Within one task, the switch times between different options varied significantly before and after training. As shown in Figure 5a, switch times before training (M = 37.47, SD = 17.38) was higher than that after training (M = 19.42, SD = 90.04). Additionally, there was a notable difference between them (95% CI: −26.11 to −10.00, p = 0.000). The t-test of switch times per second between different options also indicated a notable decrease after training (95% CI: −0.04 to −0.01, p = 0.000).

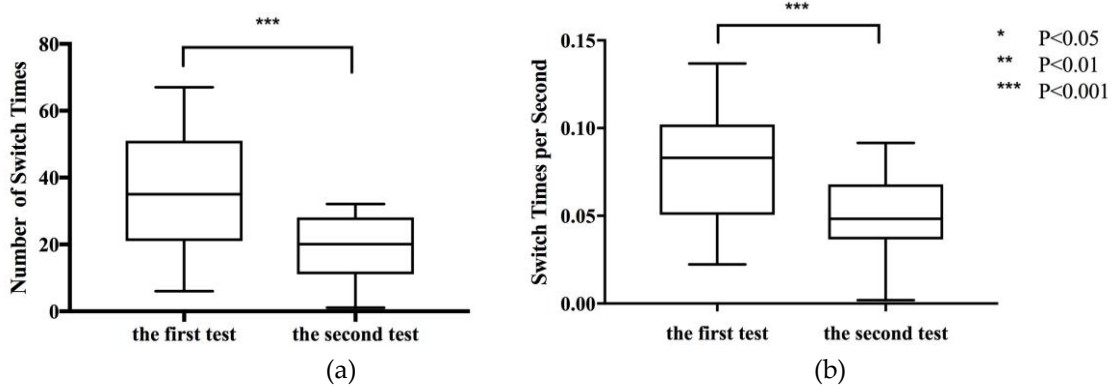

**Figure 5.** Statistics of switch times and switch times per second for different options before and after training and the results of the significance test. (**a**) switch times (18.05 ± 16.71, t = 4.708, p = 0.000); (**b**) switch times per second (0.03 ± 0.03, t = 4.299, p = 0.000).

### 3.4. Visual Attention

We made HexBin plots of three time periods for each task: Time period 1 goes from 0 s to 5 s, time period 2 goes from 5 s to 10 s and time period 3 goes from 10 s to 15 s. We chose the first fifteen seconds of data for analysis because it can reflect most of the reading strategies of the participants. See Figure 6.

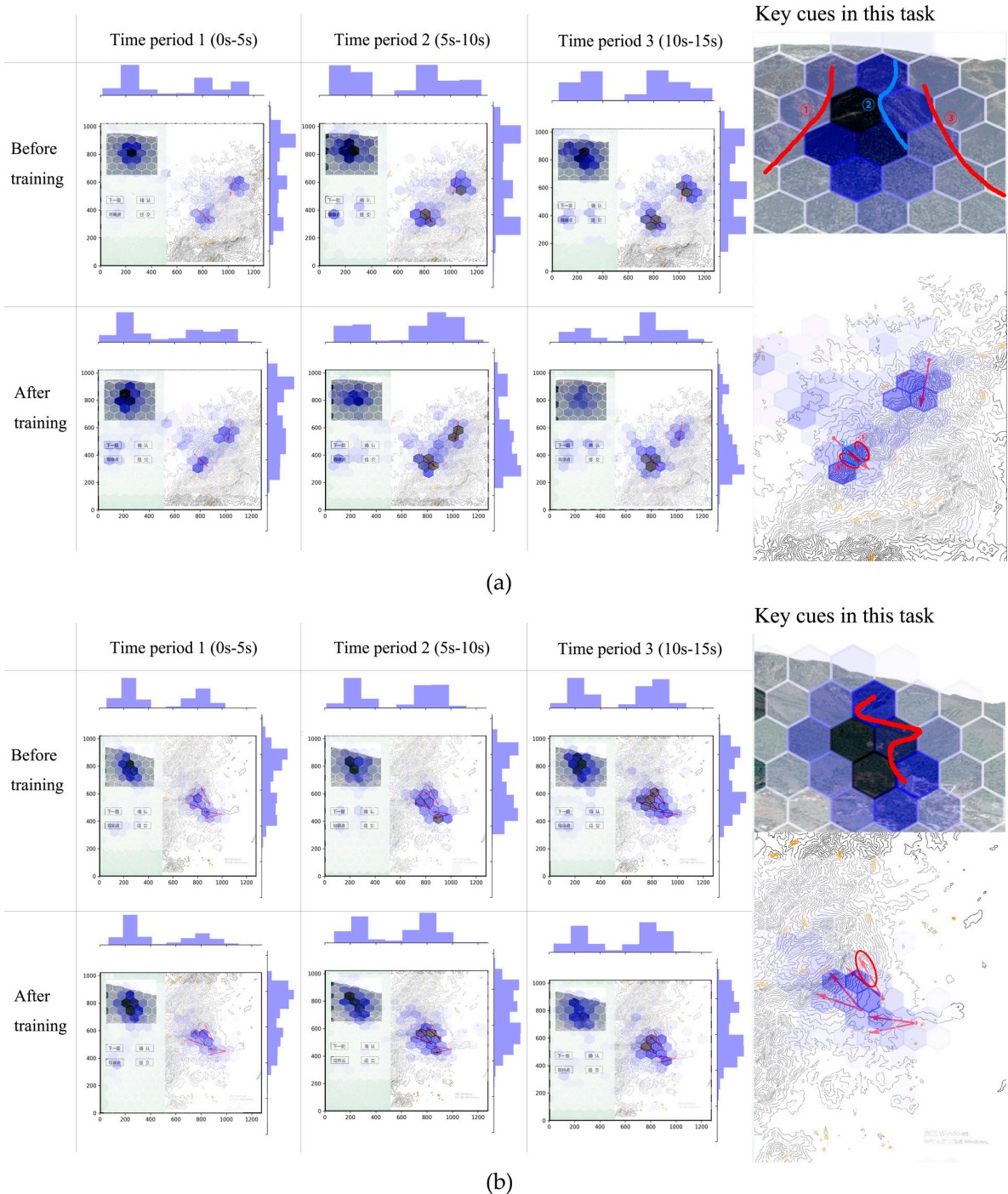

(a)

(b)

**Figure 6.** Comparison of visual attention in three periods between the first test and the second test (taking task 3 and task 4 as an example). The darkness of the colour indicates the fixation count in a hexagon; the darkest blue indicates the highest values. (**a**) Task 3 and (**b**) Task 4. The horizontal bar chart shows how the fixation count varies with the horizontal coordinates of the screen coordinates, while the vertical one reflects how the fixation count varied with the vertical coordinates.

According to Figure 6, the fixation points after training showed an obvious gathering trend in the key cues (landforms such as valleys and ridges that can be used to determine the location of the photographed scene; lines or ellipses in blue or red are indicated in Figure 6) from time period 1 to time period 3. In some tasks, there were multiple gathering centres of fixation points near the candidate answers and gathering centres tended to move to the key cues over time. In contrast, the gathering

trend towards the key cues was not obvious in the test before training. Take Task 3 for example, fixation points were concentrated in different gathering centres and the gathering centre had no obvious tendency towards the key cues from time period 1 to time period 3 (Figure 6a). For some tasks, fixation points gathered in the key cues both before and after training, but the gathering speed was faster after training. The gathering process was completed in time period 3 in the test before training, while it was completed in period 2 in the test after training (Figure 6b). To summarize, after training, fixation points were more concentrated, and specifically, the number of dark patches decreased (i.e., the number of gathering centres decreased). Fixation was more dispersed before training.

It is also worth noting that, at the beginning of a task in time period 1, the range of colour patches was larger after training (Figure 6a,b), which means that the search range was larger at time period 1.

As for the photographed scenes, fixation was concentrated in the information area in time period 1 or time period 2 after training. In time period 3, the concentration of fixation was worse and fixation count decreased; the distribution was sparse and more scattered (Figure 6a,b). Before training, fixation tended to be concentrated throughout the reading period, and the gathering centre changed slightly (Figure 6a,b). However, not all tasks showed the aforementioned characteristics. Task 5, task 7 and task 9 were different from the others. (Figure S11)

## 4. Discussion

### 4.1. Performance Analysis before and after Training

Our study concluded that the effectiveness and efficiency of the students after a series of geography courses increased. This result is expected according to related studies of geographers and non-geographers. Geographers benefit from geography education, including geography courses in college, and thus have more skills and knowledge to apply to geography problems that include reading maps and learning spatial skills. This explanation also applies to our study. The key difference between our study and research considering geographers and non-geographers is that we studied the same group of people before and after training, which can eliminate the impact of differences in individual ability with a limited number of samples. However, other variables, such as age or other courses, during the time between the tests may affect the abilities of the participants. According to the effect factors of spatial ability, we assume the slight change in age does not affect map-based abilities.

In regard to eye-movement behaviour, for indicators in processing measures, the total fixation count and total fixation duration of the photographed scenes and the topographic maps showed a decline in the mean value after training. Specifically, the total fixation duration of the topographic map showed a notable decrease after training. Fixation duration is a measure of processing depth; longer duration indicates more profound local information processing [43]. This result indicated that the amount of information processing in topographic maps decreased; before training, participants needed more time to process the (visual) information in the topographic maps.

In terms of matching measures, switch times were significantly reduced after training, which indicated that the participants were significantly less hesitant about the alternate points in the topographic map. Furthermore, the significant difference of switch times per second confirmed their decrease in hesitation after training. Additionally, both the switch times and switch times per second of the topographic map and the photographic scene increased significantly, which means that information search efficiency improved. The participants could more easily identify the area around the correct answer in the topographic map and focus on matching the information in the topographic map to that in the scene.

### 4.2. Spatial Cognitive Process Difference

Based on the above experimental results, we might be able to get an insight into the participants' spatial cognitive process and information search strategy. After training, the participants tended to scan the topographic map globally at first, and then looked for key cues, such as ridges, valleys and

hilltops, as shown in Figure 7. After that, they performed a repeated comparison of such key cues on the photographed scene and those on the topographic map, and finally made a choice. Additionally, they optimized the extraction and memorization of the key information of the photographed scenes.

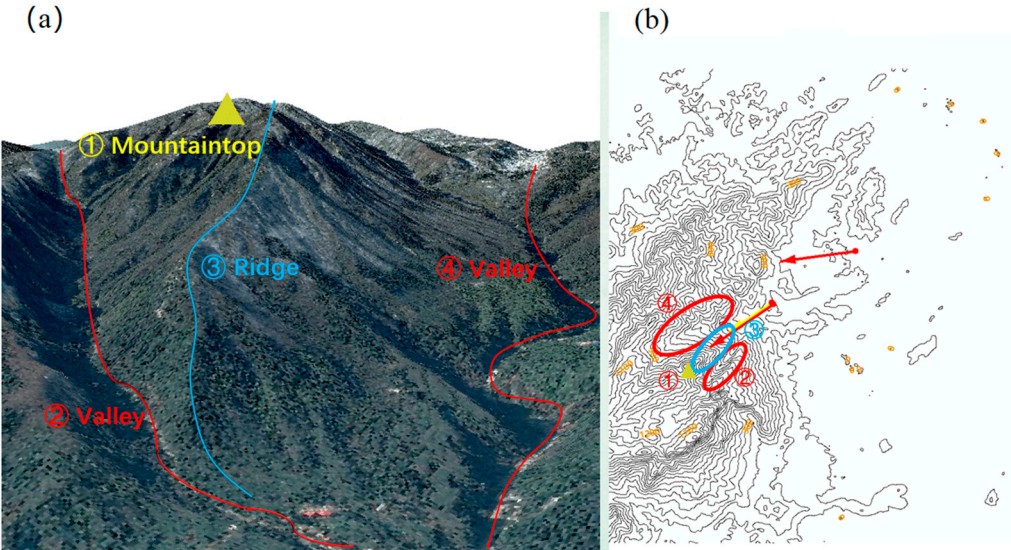

**Figure 7.** Key cues in the photographed scene (**a**) and the topographic map (**b**).

Specifically, after training, just after the participants entered the answer of each task, they scanned a larger area of the topographic map in Time period 1, which is fully reflected in Figure 6. After training, the participants were more inclined to think globally, and their attention was no longer limited to a few arrows on the candidate points. Moreover, the contour lines helped them distinguish different kinds of terrain, such as plateaus, mountains, basins, hills and plains. The increased fixating enabled them to acquire the information they wanted in topographic maps, such as the distribution of the alternate answers and the terrain.

When the trained participants completed the process above, they turned to the small-scale target for detailed information. This change after training means that the participants were able to pay more attention to key cues related to the desired topics and reduce the browsing of useless information without a goal. When their fixation was located on key cues on the topographic maps, the participants could tell the steepness of the slope using the density of the contour line and distinguish different parts of the mountain, such as peaks, ridges and valleys, from the morphological features and elevation of the contour lines. This progress in interpreting contour lines can be attributed to the repeated training of topographic map reading and drawing in the course of Physical Geography. Cross et al. also proved that students of a traditional course of instruction in map interpretation increased their ability to visualize the real-world appearance of a landform portrayed with contour lines [44]. After training, participants in this experiment obtained valid information from the key cues in the topographic maps faster and more accurately. Then, they matched the map with the key cues in the photographed scene one by one. When matching was successful, the participants tended to further search for valid information and matched the photographed scene multiple times.

Such information search and matching processes also require participants to search and process information in the photographed scenes. In this regard, participants benefitted from field trips. The main aim of field training is to teach learners to identify and observe landscape components as well as the interactions between them [45]. During the 6-month training, a long-distance field practice exercise and several short-distance field practice exercises familiarized participants with the field environment and improved their ability to imagine the photograph and extract the terrain information faster and more intuitively. The HexBin plots in Figure 6 show that the participants could extract

information from the photographed scene faster after training and photographed scene information was obtained by time period 1 and time period 2. Before training, the participants took a long time to obtain information by continuously observing the photographed scene.

### 4.3. Task Analysis

The most obvious difference between the 10 tasks is that they vary in difficulty. The spatial relationships and combinations in the photographed scene for some tasks were more complicated, with multiple tops, ridges, and valleys and other landforms that could have been used to orient the spatial information. As shown in Table 3, the accuracies before training of these tasks were low, especially for Task 7 and Task 9, which had a far lower accuracy than the other tasks, indicating that they were relatively difficult. After training, although the accuracy of the whole test was improved, the accuracies of some tasks were still very low (Task6, 8, 10). Even though participants had been trained for 6 months, the knowledge and skills learned during the training did not lead to the majority of the participants successfully solving these tasks.

By comparing the HexBin plots of each task, we found that participants' performances in Task 5, Task 7 and Task 9 (Figure S11) were significantly different from those in other tasks. Among these 3 tasks, the darkness and distribution of the coloured hexagons showed no significant difference before and after training. It indicated that participants' visual attention on the candidate points in the topographic map barely changed. And the gathering centre still had no obvious tendency towards the key cues. As mentioned by Goldstein in 2011, experts are considered to be slower thinkers because they tend to comprehend the problem instead of solving the problem [46]. After training, participants had acquired knowledge of contour lines and topographic maps; thus, they tended to think and explore more, they paid more attention to comparing the differences between the candidate options. This may also explain why the accuracies of these tasks were lower before training but increased to more than 0.50 after training.

**Table 3.** The accuracies of 10 tasks.

|                     | Task1 | Task2 | Task3 | Task4 | Task5 | Task6 | Task7 | Task8 | Task9 | Task10 |
|---------------------|-------|-------|-------|-------|-------|-------|-------|-------|-------|--------|
| **Before training** | 0.86  | 0.68  | 0.44  | 0.44  | 0.44  | 0.26  | 0.10  | 0.52  | 0.12  | 0.16   |
| **After training**  | 0.88  | 0.92  | 0.90  | 0.42  | 0.60  | 0.22  | 0.52  | 0.10  | 0.78  | 0.28   |

### 4.4. Limitations

In the two tests before and after the training, there was no change in the content of the 10 tasks. In other words, we conducted repeated tests on participants. The repeated test effect refers to the fact that the presence of previous tests may enhance and improve the performance of subsequent tests [47]. However, there is controversy about it. Some studies reached the opposite conclusion. Henkel and Blanchard-Fields believe that as the number of repeated tests increases, the increasing number of items in test recalled by the subjects also increases the effect of false memory [48]. Some studies have concluded that repeated tests did not weaken but increased the subject's susceptibility to misleading information [49]. Although there were six months between the two tests, we could not rule out the effect of the participants' memory about the test. Participants' improved performance on the second test may be due to the effect of repeated tests. However, there is no general agreement on the effect of repeated tests. Based on the data we have, it was hard for us to make a completely accurate evaluation of the repeated test effect in this experiment.

During the six months, participants also took other required courses. Among them, courses that may influence their spatial ability included Advanced Mathematics and College Physics. Many introductory college-level science courses have the potential to influence spatial ability. Pallrand and Seeber found that taking introductory physics improves visual-spatial abilities [50]. It is commonly recognized that spatial ability is positively related to mathematic skills [51], and there is correlation between the spatial rotation ability with mathematic skills [52]. So, it is not rigorous to exclude the effects of College Physics

and Advanced Mathematics on participants. It is likely that their improvement in map-based spatial ability is related to their improved mathematical skills through these two courses. In our experiment, students of the geography specialty are required to take Advanced Mathematics and College Physics as well as major required courses. Due to the constraints of practical conditions, we cannot strictly separate the influence of the former two courses from geography courses on the participants.

## 5. Conclusions

This study used an indoor eye-tracking experiment to quantitatively evaluate the changes in the map-based spatial ability of undergraduates by comparing two tests. The two tests were separated by geography courses that lasted six months. The results suggest that geography courses can improve the map-based spatial ability of undergraduates and help them complete spatial orientation and visualization tasks more quickly and accurately. Moreover, we analysed participants' strategies to solve map-based spatial problems before and after geography courses, which also verifies the effectiveness of geography courses. The results discovered in our study would direct educators to find a reliable way of improving people's spatial ability and enhance the ability to think out solutions for social and environmental problems with spatial thinking.

However, there is a lack of control over other factors related to the map-based spatial ability, such as mathematics skills, and it is difficult to strictly exclude the interference of them using the results of this study; the impact of a single course on map-based spatial ability is still unknown. Therefore, further experiments are needed to repeat the tests for more specialized courses and explore the impact of a single course. Additionally, more rigorous experiments are needed to eliminate the effects of other factors mentioned in Section 4.4.

**Supplementary Materials:** The following are available online at http://www.mdpi.com/2071-1050/11/1/76/s1.

**Author Contributions:** Conceptualization: W.D.; Methodology: Q.Y.; Formal analysis: Y.Y.; Writing and original draft preparation: S.T.; Writing/review and editing: W.D., Z.Z., B.L. and L.M.; Visualization: Q.Y.; Supervision: W.D.

**Funding:** This research is funded by the Natural Science Foundation of China (NSFC, Grant No. 41871366).

**Acknowledgments:** The authors would like to thank the reviewers for their helpful comments and suggestions.

**Conflicts of Interest:** The authors declare no conflicts of interest.

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
