# Peer review of "Using Eye Tracking to Explore the Impacts of Geography Courses on Map-based Spatial Ability"

_sustainability, doi:10.3390/su11010076_

Round 1
Reviewer 1 Report
The paper “Using Eye Tracking to Explore the Impacts of Geography Courses on Map-based Spatial Ability” describes an eye-tracking study on 52 participants. Participants performed an experiment focused on the imagination of the terrain from contours. After that they attend a 6 months geography course and performed a test again.
At the beginning, I have to say that I like the study, but there are some very serious flaws in the paper.
Abstract: At the end of the abstract, 3D photographs and 2D topographic maps are mentioned. But the purpose of the experiment and description of the task is not mentioned before – so it is confusing for the reader to understand the purpose of 2D and 3D…
Introduction is quite well…
I do not understand sentence: “According to Yoon, undergraduate students who major in science, technology, engineering and mathematics (STEM) outperformed their counterparts in terms of spatial ability when other conditions were equivalent” – who exactly are counterparts? Students of different fields?
Authors mentioned several tests (mental rotation, sense of direction…) but none of these tests was used in the study. Why? It will be extremely interesting if the performance in these tests changed after geography course as well as performance in the eye-tracking experiment.
Did authors considered also the effect of cognitive style? For example analytics and holistics like in the paper from Opach et al.
(Opach, T.; Popelka, S.; Dolezalova, J.; Rod, J.K. Star and Polyline Glyphs in a Grid Plot and on a Map Display: Which Perform Better? Cartography and Geographic Information Science 2017, 10.1080/15230406.2017.1364169, 1-20, doi:10.1080/15230406.2017.1364169.)
In my opinion nice example of expert-novice study is a paper from Burian et al.
(Burian, J.; Popelka, S.; Beitlova, M. Evaluation of the Cartographical Quality of Urban Plans by Eye-Tracking. ISPRS International Journal of Geo-Information 2018, 7, 192.)
Methods description might be better…it was quite hard to understand the design of the experiment.
“The participant sample rates (calculated by dividing the eye-tracking samples that were correctly identified by the number of attempts) of the two tests were above 70%.” – does it mean the same like Tracking Ratio? So data loss was maximally 30%?
What about the calibration results? How many participants you have to remove from the experiment?
Figure 1 – what are the orange spots?
I am not sure if I understand well the “candidate” button – so the participants click to this button or click directly into the map?
The link to the experiment might be attached to the paper…
“participants did not remember the tasks after six months” – are you sure? Did you ask them? In my opinion the way how you did the experiment is OK, but the claim that “they did not remember it” is little bit sharp.
I really hate abbreviations like RT, TFC etc…it is quite hard to orient in the text.
Regarding Score (S) – so there is one point for each task (maximum 10 points)? Or i.e. in task 6 tree points for task? This might be explained more…
HexBin plots – you mentioned Figure A1 to Figure A10 – but there is no such figure!
“TFC refers to the TFC in the area of interest (AOI), which represents the importance of an area.” – what do you mean by AOI? Area of Interest around map an picture? Or AOI around particular element in map?
“A higher fixation means more attention was given to that area.” – do you mean fixation count?
You used paired t-test – so data had normal distribution? In other case you had to use nonparametric variant of t-test (Wilcoxon test)
IMPORTANT: Figure 2 and 3 – Results are displayed for the whole experiment? It will be interesting to show us results also for each task separately…
IMPORTANT: I miss the interpretation of the results – for example - what does it mean that ST was higher?
IMPORTANT: Very problematic part of the paper is the description of HexBins – how they were created? What is in Figure 6? Upper part shows number of fixations in the 3D image and lower part in the map? Are the colour scales same in all examples? For example if dark one means more than X fixations? Why there is no legend?
According to Table 3 – THERE IS NO TABLE 3 IN THE PAPER
“We chose the first fifteen seconds of data for analysis because it can reflect most of the reading strategies of the participants.” – source?
Chapter 4.2. It will be nice to prove the claims with some data – for example visualization using sequence chart or some scanpath comparison?
As I said at the beginning – I really like the study, but the paper needs major revisions, especially in the part of data analysis. I am also surprised that I did not found figure Figure A1 to Figure A10 and Table 3. To be honest, when I found these errors, I thought about immediate rejection – because authors took the paper back for “further improve the paper” – and there are so huge problems like missing figures and tables.
I will advise to be more rigorous during the work on the future papers.
Author Response
Dear reviewer,
thanks for you comments and advice, we have revised our manuscript. The point-by-point response is attached, please check.

Reviewer 2 Report
Introduction.
The authors present a throughout introduction to spatial abilities and geography courses. However, the eye-tracking part of the introduction seems a bit small and relevant information are missing. I would suggest adding some explanations about eye-tracking (how does it work, what are the main measured parameters) for readers who are unfamiliar with the technique. A good example should be:
Holmqvist, K., Nyströom, M., Andersson, R., & van de Weijer, J. (2011). Eyetracking. A comprehensive guide to methods and measures. Oxford: Oxford University Press.
The aims are well-defined and clear.
Materials and methods.
The authors use a T120 device but apply 60Hz during the measurements. Could you please clarify why did you choose lower rates when 120Hz could have been also used?
Were the tasks randomized between participants and sessions? Randomization is essential to avoid order effect.
Again, for readers who are unfamiliar with eye-tracking, the terms “fixation” and “AOI” would be hard to understand.
AOI is not defined in table 2.
Results.
Did the authors have normally distributed data to use paired t-tests?
When introducing the results presented on Figure 3a-f, the authors say higher and lower values but statistical comparison is done only in the case of Figure 3d. Are the other results non-significant or did you miss mentioning these? If the results are not significant, you need to present that although the means are higher, the difference cannot be justified by statistical tests.
Visual attention.
Could you please explain more this sentence:
“We chose the first fifteen seconds of data for analysis because it can reflect most of the reading strategies of the participants.” Why fifteen? Why not 12 or 17?
I was unable to find Figure A11
In summary, the presented article is a valuable work, it has a clear golden line a nice, readable style.
Author Response
Dear reviewer,
Thanks for you comments and advice. We have revised our manuscript. The point-by-point response is attached, please check.

Reviewer 3 Report
The manuscript by Dong et al. used eye-tracking technology to provide an intra-individual quantitative assessment of the impact of geography courses in map-based spatial ability of undergraduate students. Through this method they were able to determine performance accuracy and infer on the cognitive process to solve tasks that required map-based spatial localization. The methods are sound and the results are properly presented with conclusions well drowned, except for a few recommendations and questions this reviewer would like to be better clarified.
1 – Last paragraph of introduction is not necessary as it does not present any essential information needed to comprehend the study or the manuscript. Moreover, sections are incorrectly identified: section 1 is research background, section 2 is methods, and so on.
2 – It is recommended to explicitly outline what the overall goal of the study is. In the first paragraph on page 4, the authors only describe their procedures to evaluate the impact of geography courses on spatial abilities. The goal is somewhat inferred by entire reading of the manuscript, but because the authors also discuss the effectiveness of eye-tracking as a critical method for this type of analysis, it is sometimes misunderstood what is the most relevant direction of the study - whether to demonstrate importance of geography courses or to demonstrate eye-tracking as important technology for these assessments.
3 – Section 2.1: Describe what were inclusion and exclusion criteria used to enroll participants and if they signed informed and research authorization consent. Inform if the research was approved by an ethics committee. Also, comment about possible other courses the participants were taking in addition to the geography courses outlined.
4 – Section 2.4: The authors stated “…the participants did not remember the tasks after six months.” How the authors can ensure that the participants had no recollection of the tasks? This variable cannot be completely ruled out without proper examination. Hence, this should be considered an important factor influencing results of the tests after training and should be properly discussed.
5 – Why the photograph and topographic maps as a whole were considered areas of interest (AOI) for subsequent analysis if it was possible to list several AOI by taking each hexagon as one AOI, as done for visual memory assessment?
6 - It is necessary to add a section to describe in details the statistical analysis conducted: if data was parametric or non-parametric and the adequate statistical tests used for each analysis/comparison.
7 - Section 3.1: Are values for RT before and after training correct? The authors say that t-test revealed a significant DECREASE after training (and this is also seen in the Figure 2b), but the values show the opposite.
8 - Please provide a description of statistical methods and data representation (mean ± SD, N , p, etc.) in each figure caption.
9 - Page 9, last paragraph: “whereas the AFD for the fixation duration…” do you mean “topographic map”?
10 - Page 12: There is no table 3 presented in the manuscript.
11 - Page 12: The authors state “…there were no significant differences in the fixation point distributions…”. However, the statistical analysis for this data is not described anywhere in the manuscript. Please provide this information either in the methods or results section.
12 - Page 13, end of second paragraph: there is no figure A11 presented in the manuscript.
13 - Discussion, first paragraph: “Students know little about geography before they take geography courses; thus, they may GET anxious when given a geography task and proceed cautiously.” This sentence is very speculative without proper scientific demonstration of its veracity.
14 - “However, other variables during the time between the tests may affect the abilities of the participants.” Like what? Please provide a list of important variables to consider (other courses, personal field practice…)
15 - Figure 7: number 1 in yellow identifies the mountaintop in the photograph. It should also be colored yellow in the topographic map to maintain the pattern.
16 - Section 4.3: the authors discuss the performance of the participants in tasks 5, 7 and 9 and state that it was significantly different from their performances in other tasks. However, they do not show any score or RT values neither demonstrate the statistical tests conducted. Please include this in the results section for appropriate discussion. Also, there is a typo in the next sentence: have HAD.
17 - Page 16, first line: missing word: THAN.
18 - In the conclusion, the authors mention the possibility of participants taking other courses and that this could influence spatial ability as well. Because this is a very important factor, there should be properly discussed as a limitation of the study (and outline others limitations as pointed by this reviewer) in the discussion section, not only in the conclusion.
19 - Further should be in lower case.
“…more rigorous experiments are needed to eliminate the effects of other factors.” Like what (see comment 14)?
Author Response

(The authors gave the same response as above.)

Round 2
Reviewer 1 Report
Authors implemented all my remarks and now I am satisfied with the paper and agree with its publication.